# Revisiting the Transferability of Few-Shot Image Classification: A Frequency Spectrum Perspective

**DOI:** 10.3390/e26060473

**Published:** 2024-05-29

**Authors:** Min Zhang, Zhitao Wang, Donglin Wang

**Affiliations:** 1College of Computer Science & Technology, Zhejiang University, Hangzhou 310027, China; zhangmin@westlake.edu.cn; 2School of Engineer, Westlake Univercity, Hangzhou 310030, China; wangzhitao@westlake.edu.cn

**Keywords:** few-shot learning, few-shot image classification, frequency spectrum, distribution shift, causal perspective

## Abstract

Few-shot learning, especially few-shot image classification (FSIC), endeavors to recognize new categories using only a handful of labeled images by transferring knowledge from a model trained on base categories. Despite numerous efforts to address the challenge of deficient transferability caused by the distribution shift between the base and new classes, the fundamental principles remain a subject of debate. In this paper, we elucidate **why** a decline in performance occurs and **what** information is transferred during the testing phase, examining it from a frequency spectrum perspective. Specifically, we adopt causality on the frequency space for FSIC. With our causal assumption, non-causal frequencies (e.g., background knowledge) act as confounders between causal frequencies (e.g., object information) and predictions. Our experimental results reveal that different frequency components represent distinct semantics, and non-causal frequencies adversely affect transferability, resulting in suboptimal performance. Subsequently, we suggest a straightforward but potent approach, namely the ***Fr**equency **S**pectrum **M**ask* (FRSM), to weight the frequency and mitigate the impact of non-causal frequencies. Extensive experiments demonstrate that the proposed FRSM method significantly enhanced the transferability of the FSIC model across nine testing datasets.

## 1. Introduction

Few-shot image classification (FSIC) endeavors to identify unlabeled images within the query set belonging to novel classes, leveraging knowledge acquired from base classes with a few labeled images in the support set [1,2]. Recently, numerous methodologies have emerged to address the challenge of few-shot image classification, which are mainly divided as follows. (1) **Fine-tuning-based methods** [3,4,5] tackle the problem by *learning to transfer*. They follow the standard machine learning or transfer learning [6,7,8] procedure to pretrain transferable knowledge and test-tune the knowledge in FSIC episodes sampled from novel classes. (2) **Metric-based methods** [9,10] solve the problem by *learning to compare*. They calculate the similarity between the query with the unlabeled data and the support set with labeled images. Finally, (3) **meta-based methods** [11,12,13] address the problem by *learning to learn*. They learn a good model initialization for fast adaptation to novel classes.

When distribution shifts exist between the base (or training dataset) and novel classes (or testing dataset), it is common for the transferability of the model trained on the base classes to diminish. In Figure 1, we scrutinize the reasons behind performance declines resulting from distribution shifts by employing T-SNE [14] visualization. This analysis is conducted using Resnet12 trained on the *mini*ImageNet-train dataset. In Figure 1a–d, we can observe that Figure 1a demonstrates superior class clustering performance. This can be attributed to the training model using a training dataset that closely resembled the testing dataset during the training phase.

Recently, the majority of works have been dedicated to addressing the issue of distribution shift [15,16,17]. However, these works predominantly concentrated on **enhancing transferability** by introducing diverse regularizations or larger model architectures, often leading to increased memory and time consumption in exchange for marginal performance improvements [18,19]. The primary challenge of these works stems from a lack of clarity regarding **why** FSIC performance diminishes and **what** information undergoes transferring during the distribution shifts. In this paper, we seek to elucidate the underlying mechanism behind the remarkable transferability observed in the FSIC problem. Therefore, we approached the problem by constructing a causal graph of FSIC from a frequency spectrum perspective, motivated by both theoretical and experimental evidence suggesting that the frequency space contains more distinguishable semantic information than the feature space, as established in Figure 2 and prior studies [20,21]. Figure 2 showcases the average amplitudes of the eigenfrequencies across four testing datasets using a pretrained model, vividly illustrating pronounced distinctions in the frequency space among testing datasets.

In this paper, we construct a causal graph of FSIC within the frequency space which delineates the relationships among the causal frequency (e.g., object information), non-causal frequency (e.g., background knowledge), and prediction. As illustrated in Figure 3a, our causal assumption posits the non-causal frequency as a confounder between the causal frequency and prediction [22]. For example, consider a scenario where the dog images in the training data feature grass backgrounds. This scenario poses a challenge during the testing phase when encountering the dog image with a water background, as the inconsistent background information hampers recognition. Consequently, the presence of **confounding information stands as a primary factor** influencing the performance of FSIC under distribution shifts. Notably, the **causal frequency, representing domain-invariant knowledge, should be effectively transferred** from training to testing datasets. We introduce a straightforward yet potent method called the **Fr**equency **S**pectrum **M**ask (FRSM) to weight the influences of causal frequency components. Through empirical validation, we benchmark the FRSM against state-of-the-art (SOTA) methods across nine FSIC datasets, demonstrating its efficacy. Our primary contributions can be outlined as follows:We clarify **why** there is a decline in performance and **what** information is transferred during the distribution shift in the few-shot image classification task from a frequency spectrum perspective.We adopt a causal perspective of few-shot image classification to demonstrate that non-causal frequencies impact transferability and introduce a straightforward yet efficient method, the FRSM, to weight frequencies.The experimental results indicate that the proposed FRSM method achieves superior performance compared with representative state-of-the-art methods in the few-shot image classification task.

## 2. Related Work

### 2.1. Few-Shot Image Classification

Most works have been proposed in recent years to solve the FSIC problem. These methods are broadly categorized into three groups: (1) *fine-tuning-based methods* employ a non-episodic paradigm to pretrain the model on base classes and then test-tune the pretrained model on novel classes. For example, Baseline and Bseline++ [3] propose using the fully connected layer or cosine distance to calculate the classification loss, respectively. SKD [23] further employs rotation-based self-supervision learning during pretraining to enhance the feature extraction process. (2) *Metric-based methods* aim to learn a discriminative embedding space by leveraging a learned distance metric. For example, ProtoNet [10] proposes performing classification by measuring the Euclidean distance between the query and prototype representation of each class. RelationNet [24] proposes calculating the distance between the query and prototype by leveraging the relational module. (3) *Meta-based methods* focus on learning how to optimize a model through bi-level optimization. In particular, MAML [11] proposes quickly adapting to novel classes with only a few labeled images and a small number of gradient updates. LEO [12] proposes learning a stochastic latent space from high-dimensional parameters. Our method tends toward fine-tuning based methods due to their simplicity and effectiveness.

### 2.2. Frequency Spectrum Learning

In conventional image processing, frequency analysis has been widely studied for years. Recently, it has been set forth to be incorporated into deep learning methods. Some researchers have been working on analyzing and understanding some behaviors of deep neural networks. The frequency spectrum has garnered considerable attention from researchers due to its utility in analyzing and comprehending the behavior and interpretability of deep neural networks. For example, in [25], the authors observed that convolutional neural networks (CNNs) exhibit a pronounced bias toward recognizing textures rather than shapes. Some research efforts have aimed at enhancing the generalization capability of CNNs by leveraging insights from the frequency spectrum. Notably, FACT [26] proposes forcing the model to capture phase information under the assumption that phase information is more robust to distribution shifts. FSDR [27] improves the generalization capability through randomizing images in a frequency space, preserving domain-invariant frequency components while randomizing domain-variant ones exclusively. Diverging from prior research, ours centers on analyzing the transferability of the FSIC problem across base and novel classes through frequency analysis.

## 3. Methodology

In this section, we begin by delineating the problem formulation of few-shot image classification, setting the stage for a detailed exploration of this challenge. Following this, we adopt a novel approach by examining the issue through the lens of causal analysis from a frequency spectrum perspective. This allows us to identify key frequency components that significantly influence classification outcomes, especially when limited data are available. Lastly, we introduce a straightforward yet potent technique: the frequency spectrum mask (FRSM). This method strategically adjusts the weights of frequency components to enhance the model’s capability to emphasize the most pertinent features for classification. This refinement ultimately leads to improved accuracy and robustness in few-shot learning scenarios. By implementing the FRSM, we aim to mitigate the impact of irrelevant frequency noise and amplify the contribution of crucial frequencies.

### 3.1. Few-Shot Image Classification

Several approaches have been proposed to tackle the few-shot image classification (FSIC) problem. Among these, fine-tuning-based methods have emerged as particularly effective due to their simplicity and notable efficacy. Therefore, we developed the FRSM based on fine-tuning techniques. These methods typically adhere to the standard transfer learning procedure, which comprises two phases: the **pretraining phase**, using the training set $D_{train}$, and the **test-tuning phase**, employing a testing set $D_{test}$. Additionally, a validation set $D_{val}$ is utilized for model selection during the test-tuning phase.

**During the pretraining phase**, the entire training set $D_{train}$ is employed to pretrain the feature extractor and classifier on base classes $C_{base}$ using the standard cross-entropy loss $L_{ce}$ as follows:(1)$$\underset{\theta ,\omega }{\arg\min}\sum_{i=1}^{N}L_{ce}\left(g_{\omega }\left(z_{i}\right),y_{i}\right),\;\;s.t.\;z_{i}=f_{\theta }\left(x_{i}\right),$$
where *f* and *g* are the feature extractor and classifier, parameterized by θ and ω, respectively. The pairs $\left\{x_{i},y_{i}\right\}$ represent images and their corresponding labels from the training set, denoted as $D_{train}={\left\{x_{i},y_{i}\right\}}_{i=1}^{N}$, where $x_{i}\in R^{3\times H\times W}$, *N* is the total number of images and *H* and *W* represent the height and width of the images, respectively, with the one-hot labeling vector $y_{i}\in Y=\left\{0,1\right\}$.

**During the test-tuning phase**, we sampled a C-way K-shot episode from the testing set $D_{test}$ containing novel classes $C_{novel}$. Each episode T comprises a support set S and a query set Q, denoted as $T=\langle S,Q\rangle$, where $S={\left\{x_{i},y_{i}\right\}}_{i=1}^{CK}$ represents the support samples and $Q={\left\{x_{i},y_{i}\right\}}_{i=1}^{CM}$ denotes the query samples, with CK denoting the number of support samples and CM denoting the number of query samples. A new C-class classifier will be relearned based on S every time. Basically, the pretrained embedding parameter θ is fixed to avoid overfitting because there are limited labeled data in S. Once the novel classifier is learned, the labels of Q can be predicted.

### 3.2. A Causal Graph of FSIC

We used a casual graph of the FSIC to model and construct a structural causal model (SCM) in Figure 3a. It shows the causality relationship among five variances: the input data *X*, non-causal frequency $F_{1}$, causal frequency $F_{2}$, frequency *F*, and prediction or label *Y*, where the link from one variable to another indicates the cause-and-effect relationship (cause → effect). We will now list the following explanations for our proposed SCM:**$F_{2}\leftarrow X\to F_{1}$.** The variable $F_{2}$ signifies the causal frequency that genuinely represents the inherent characteristic of the input data *X*, such as the object details in an image. Conversely, $F_{1}$ indicates the non-causal frequency typically resulting from biases in the data or superficial patterns, such as the background details of an image. Given that $F_{2}$ and $F_{1}$ coexist within the input data *X*, these causal relationships are established.**$F_{2}\to F\leftarrow F_{1}$.** The variable *F* corresponds to the frequency of the input data *X*; that is, $F=FFT(f_{\theta }\left(X\right))$, with FFT denoting the **f**ast **F**ourier **t**ransformation. To produce *F*, the traditional learning approach uses both the non-causal frequency $F_{1}$ and the causal frequency $F_{2}$ to extract the discriminative features.**F→Y.** The primary aim of learning via the frequency spectrum is to ascertain the attributes of the input data *X*. The classifier $g_{\omega }$ will determine the prediction *Y* based on the frequency *F*; specifically, $Y=g_{\omega }\left(iFFT\left(F\right)\right)$, where iFFT stands for the **i**nverse **f**ast **F**ourier **t**ransformation.

When scrutinizing our proposed SCM, we acknowledge the role of the non-causal frequency $F_{1}$ as a confounding factor between $F_{2}$ and *Y*, stemming from the existence of a backdoor path between $F_{2}$ and *Y*; specifically, $F_{2}\leftarrow X\to F_{1}\to Z\to Y$. Even if $F_{2}$ lacks a direct connection to *Y*, the presence of this backdoor path can lead $F_{2}$ to exhibit a spurious correlation with *Y*, resulting in erroneous predictions based on the non-causal frequency $F_{1}$ instead of the causal frequency $F_{2}$. Hence, it is imperative to enhance the transferability of FSIC by adjusting the frequencies and mitigating the influence of the non-causal frequency. In this paper, we propose a straightforward yet effective method, the FRSM, for assigning weights to each frequency.

### 3.3. Frequency Spectrum Mask

In this section, we introduce our FRSM to weight each frequency. Figure 3b illustrates the overall learnable process of the FRSM. Note that the FRSM is used in the test-tuning phase with novel classes $C_{novel}$, which uses a frozen feature extractor $f_{\theta }$ pretrained on the base classes $C_{base}$. The reason for this is that the distribution shift between the base and novel classes makes it impossible for a feature extractor pretrained on the base classes to extract features suitable for novel classes. Therefore, we needed to weight the extracted features to meet the requirements of the novel classes. Following previous works [20,26], the FRSM was developed in the frequency space, as the frequency space has more information than the feature space.

In the test-tuning phase, firstly, a C-way K-shot episode $T=\langle S,Q\rangle$ is sampled from the novel classes $C_{novel}$. Then, for each episode, the few labeled support images $S={\left\{x_{i},y_{i}\right\}}_{i=1}^{CK}$ are used to pre-learn a new classifier $g_{\omega }$ and our FRSM $M_{m}$ parameterized by *m*, based on the frozen feature extractor $f_{\theta }$. Finally, the unlabeled query images Q are used to evaluate the performance of the test-tuned classifier $g_{\omega }$ and FRSM $M_{m}$. Next, we give a detailed description of the test-tuning phase.

We used the frozen $f_{\theta }$ to extract the features $Z_{S}$ of each image in S (i.e., $Z_{S}$ = ${\left\{z_{i}\right\}}_{i=1}^{CK}$ with $z_{i}=f_{\theta }\left(x_{i}\right)$ and $z_{i}\in R^{C\times \bar{H}\times \bar{M}}$, where *C* is the channel and $\bar{H}$ and $\bar{M}$ are the height and width of the feature representations, respectively). For each image feature, we transformed it into frequency space to find the corresponding frequency representation $z_{i}^{F}$ with a fast Fourier transformation (FFT), which is formulated as follows:(2)$$\begin{array}{cc} z_{i}^{F}\left({\bar{h}}^{F},{\bar{w}}^{F}\right) & =FFT(z_{i}\left(\bar{h},\bar{w}\right))\\ \; & =\sum_{\bar{h}=0}^{\bar{H}-1}\sum_{\bar{w}=0}^{\bar{W}-1}z_{i}\left(\bar{h},\bar{w}\right)e^{-j2\pi ({\bar{h}}^{F}\frac{\bar{h}}{\bar{H}}+{\bar{w}}^{F}\frac{\bar{w}}{\bar{W}})}, \end{array}$$
where $\left(\bar{h},\bar{w}\right)\in \left(\bar{H},\bar{W}\right)$ is the height and width pair in the feature space and $\left({\bar{h}}^{F},{\bar{w}}^{F}\right)\in \left({\bar{H}}^{F},{\bar{W}}^{F}\right)$ is the corresponding height and width pair in the frequency spectrum space. Following the literature [26], the frequency representation $z_{i}^{F}$ includes the amplitude $z_{i}^{A}$ and phase $z_{i}^{P}$ components. Then, the two components $z_{i}^{A}$ and $z_{i}^{P}$ are calculated as follows:(3)$$\begin{array}{cc} \; & z_{i}^{A}\left({\bar{h}}^{F},{\bar{w}}^{F}\right)={\left[R^{2}\left(z_{i}\right)\left({\bar{h}}^{F},{\bar{w}}^{F}\right)+I^{2}\left(z_{i}\right)\left({\bar{h}}^{F},{\bar{w}}^{F}\right)\right]}^{1/2},\\ \; & z_{i}^{P}\left({\bar{h}}^{F},{\bar{w}}^{F}\right)=\arctan\left[\frac{I\left(z_{i}\right)\left({\bar{h}}^{F},{\bar{w}}^{F}\right)}{R\left(z_{i}\right)\left({\bar{h}}^{F},{\bar{w}}^{F}\right)}\right], \end{array}$$
where $R(z_{i})$ and $I(z_{i})$ represent the real and imaginary parts of $z_{i}^{F}\left({\bar{h}}^{F},{\bar{w}}^{F}\right)$, respectively. The operator arctan is the inverse tangent function. Since $z_{i}^{A}$ represents the magnitude of the frequency, it is crucial to assign appropriate weights to it for accurate representation and analysis. To achieve this, we propose using our proposed FRSM, denoted as $M_{m}$. By applying this method, we can effectively weight $z_{i}^{A}$, thereby enhancing the precision. The transformed frequency components are formulated as shown in the equation below:(4)$${\hat{z}}_{i}^{A}\left({\bar{h}}^{F},{\bar{w}}^{F}\right)=M_{m}⊙z_{i}^{A}\left({\bar{h}}^{F},{\bar{w}}^{F}\right),$$
where $M_{m}\in R^{C\times \bar{H}\times \bar{W}}$ is initialized to one and ⊙ is the element-wise product. The weighted amplitude ${\hat{z}}_{i}^{A}$ is combined with the original phase $z_{i}^{P}$ to form a new frequency ${\hat{z}}_{i}^{F}$:(5)$${\hat{z}}_{i}^{F}\left({\bar{h}}^{F},{\bar{w}}^{F}\right)={\hat{z}}_{i}^{A}\left({\bar{h}}^{F},{\bar{w}}^{F}\right)*e^{-j*z_{i}^{P}\left({\bar{h}}^{F},{\bar{w}}^{F}\right)}.$$

The new frequency representation ${\hat{z}}_{i}^{F}$ can be transferred to the original feature space via an inverse fast Fourier transformation (iFFT), which can be formulated as follows:(6)$$\begin{array}{cc} {\hat{z}}_{i}\left(\bar{h},\bar{w}\right) & =iFFT({\hat{z}}_{i}^{F}\left({\bar{h}}^{F},{\bar{w}}^{F}\right))\\ \; & =\frac{1}{\bar{H}\cdot \bar{W}}\sum_{\bar{h}=0}^{\bar{H}-1}\sum_{\bar{w}=0}^{\bar{W}-1}{\hat{z}}_{i}^{F}\left({\bar{h}}^{F},{\bar{w}}^{F}\right)e^{j2\pi ({\bar{h}}^{F}\frac{\bar{h}}{\bar{H}}+{\bar{w}}^{F}\frac{\bar{w}}{\bar{W}})}. \end{array}$$

The new feature representation ${\hat{z}}_{i}$ is used to replace the previous feature representation $z_{i}$ in Equation (Equation 1). Then, the cross-entropy loss $L^{ce}$ is calculated to update the new classifier $g_{\omega }$ and our proposed FRSM $M_{m}$.

## 4. Experiment

In this section, the research questions guide the running of experiments as follows. **Q1:** Why does the transferability decrease as the distribution shifts between the base and novel classes increase? **Q2:** What information should be transferred from the base classes to the novel classes? **Q3:** How effective is our FRSM for few-shot image classification tasks in all testing datasets?

### 4.1. Experimental Set-Up

**Datasets.** In our experiments, for the training dataset $D_{train}$, we selected the train split of *mini*ImageNet [9] to pretrain the feature extractor due to the diversity and complexity of its images. The training dataset was also named *mini*ImageNet-train. For the testing dataset $D_{test}$, we chose nine benchmarks, including the subsets of ILSVRC-2012 [28] (i.e., the test split of *mini*ImageNet, named *mini*ImageNet-test, and *tiered*ImageNet), all evaluation datasets of cross-domain few-shot learning (CDFSL) [29] (i.e., EuroSAT, CropDisease, ChestX, and ISIC), and the evaluation benchmarks proposed by Tseng et al. [15] (i.e., CUB, Cars and Plantae). All images in the above datasets were resized to 84×84 pixels, and data augmentation was used. For more details, please refer to LibFSL [30] and Table 1.

**Baseline.** To evaluate the effectiveness of our FRSM method, we selected seven classic and effective methods for comparison. They included Baseline [3], Baseline++ [3], and SKD [23] from the fine-tuning-based methods, ProtoNet [10] and RelationNet [24] from the metric-based methods, and MAML [11] and LEO [12] from the meta-based methods, where SKD and LEO recognized the novel classes based on the pretrained model in the training phase.

**Experimental details.** Following the literature [30], we adopted two different feature extractors: Conv64F (see Table 2) and ResNet12 (see Figure 1). In the test-tuning phase, we used the pretrained feature extractor from [23]. We utilized the SGD optimizer with a momentum of 0.9 and a weight decay of 1e−3. The learning rates for the re-learned classifier and our FRSM was initialized as 1e−2 and 3, respectively. For each example of testing data, we randomly sampled 600 episodes, and each episode contained 5 classes. Each class had 5 or 10 labeled images (support set) and an additional 15 unlabeled images (query set) for performance evaluation, formulating the 5 way 5 shot or 10 shot episode.

### 4.2. Experimental Results

In this section, we run experiments to answer the following questions.
**Q1. The transferability from base to novel classes**. The experimental results in Figure 4 aim to answer Q1. In Figure 4, we show the similarity of the testing and training datasets according to the amplitude ratio. The testing datasets’ similarity to the training dataset was ordered as follows: *mini*ImageNet-test > Cars > Plantae > EuroSAT > ISIC > ChestX. The similarity is known to affect the transferability of the training dataset features into the testing datasets. To this end, we found that the datasets’ similarity and few-shot difficulty simultaneously led to performance degradation.**Q2. The information from base to novel classes.** We conducted experiments on testing datasets, with the results shown in Figure 2. We found that the high-frequency components in different datasets had more similarity than the low-frequency components. Therefore, compared with the low-frequency components, the high-frequency components contributed to the transferability more.**Q3. The performance of the FRSM.** To answer Q3, we conducted experiments on eight datasets. From Table 2, which reports the average classification accuracy, we have the following findings. (1) Our proposed FRSM achieved the best performance in most settings. This is because our FRSM assigned a large weight to the causal frequency and a low weight to the non-causal one, thereby avoiding the influence of the confounder on the transferability. (2) Surprisingly, our FRSM was weaker than the baseline for relatively similar testing data (e.g., *tiered*ImageNet). The possible reason for this is that the confounder is generally simpler. For example, learning the backgrounds of images is easier than learning the foreground, and this can help the transferability from similar base classes to novel classes. Our FRSM suppressed the confounder and led to performance degradation. Nevertheless, we should focus more on testing datasets with larger distribution shifts, because this case is more in line with real-world environments.


## 5. Conclusions

In this paper, we explored the underlying reasons for the decline in performance and transferability encountered when facing distribution shifts between base and novel classes. We approached this investigation from the perspective of frequency spectrum analysis. Furthermore, we clarified **which** information ought to be transferred from base to novel classes. Through a causal perspective on FSIC, we delved into the phenomenon and demonstrated that non-causal frequencies can profoundly influence transferability as confounding factors. Therefore, we proposed a straightforward yet effective method, the FRSM, to dynamically weigh each frequency using a learnable paradigm. This approach helps mitigate the influence of confounders and enhances transferability. Extensive experiments demonstrated that our proposed FRSM method achieved new state-of-the-art results.

## Figures and Tables

**Figure 1 entropy-26-00473-f001:**
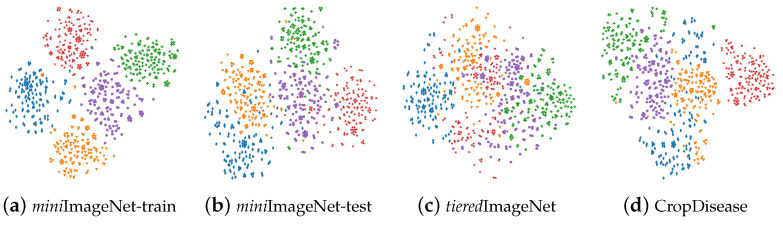
T-SNE [14] visualization of four datasets utilizing a model pretrained on the training set of *mini*ImageNet (i.e., *mini*ImageNet-train). Here, *mini*ImageNet-test means the testing data of *mini*ImageNet. From left to right, the performance of the class cluster declines, which can be attributed to the impoverished similarity between the training and testing datasets.

**Figure 2 entropy-26-00473-f002:**
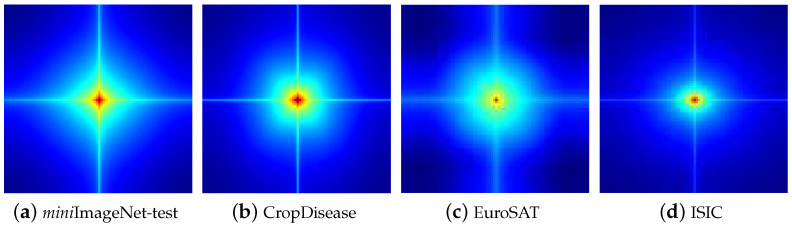
The average amplitudes of the eigenfrequencies across four datasets, applying a model pretrained on the training set of *mini*ImageNet (i.e., *mini*ImageNet-train). Here, *mini*ImageNet-test means the testing data of *mini*ImageNet. Proximity to the center signifies a low-frequency component, while being far from the center indicates a high-frequency component.

**Figure 3 entropy-26-00473-f003:**
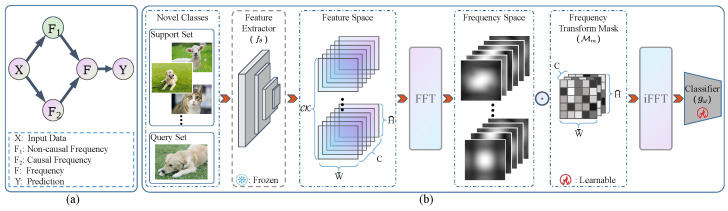
(**a**) A causal look at FSIC from the frequency spectrum perspective. (**b**) The test-tuning process. The FRSM weights the frequency learning from novel classes in the testing or fine-tuning phase, which uses a frozen feature extractor pretrained on base classes.

**Figure 4 entropy-26-00473-f004:**
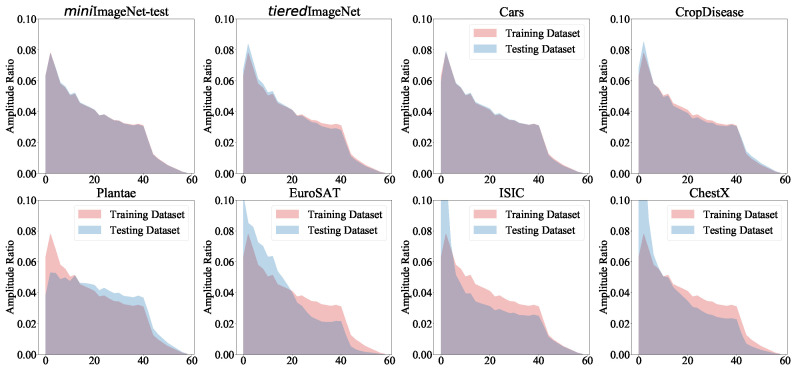
The average amplitude ratio between six testing datasets and the training dataset (i.e., *mini*ImageNet-train).

**Table 1 entropy-26-00473-t001:** Summary of testing datasets we used in this paper. For each dataset, we picked 5 classes, and we show illustrative images.

Dataset	Samples	Number of Classes	Number of Samples
*tiered*ImageNet	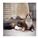	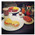	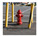	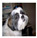	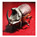	160	206,209
CUB-200-2011	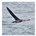	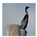	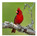	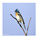	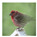	200	11,788
ISIC	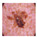	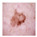	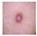	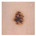	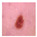	7	10,015
CropDisease	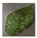	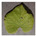	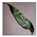	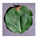	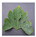	38	43,456
Cars	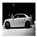	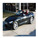	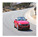	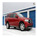	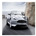	196	16,185
EuroSAT	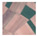	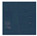	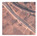	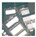	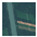	10	27,000
Plantae	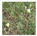	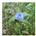	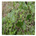	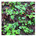	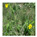	69	26,650
ChestX	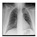	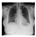	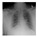	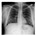	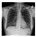	7	25,848

**Table 2 entropy-26-00473-t002:** Few-shot image classification average accuracy (%) on eight testing datasets under the 5 way 5 shot or 5 way 10 shot settings. The model was trained on *mini*ImageNet. The best results are bold.

Model	*tiered*ImageNet		CUB-200-2011		CropDisease		EuroSAT		Average	
	5 Shot	10 Shot	5 Shot	10 Shot	5 Shot	10 Shot	5 Shot	10 Shot	5 Shot	10 Shot
Baseline [3]	65.49%	71.72%	55.41%	63.80%	81.21%	87.84%	70.12%	76.99%	68.06%	75.09%
Baseline++ [3]	68.68%	72.62%	53.64%	59.41%	60.60%	68.59%	55.12%	60.09%	59.51%	65.18%
SKD [23]	66.93%	72.70%	55.59%	64.28%	80.36%	87.46%	70.85%	77.18%	68.44%	75.41%
ProtoNet [10]	**71.81%**	**76.17%**	**57.65%**	**64.34%**	80.15%	85.76%	69.03%	73.10%	69.66%	74.84%
RelationNet [24]	70.16%	64.86%	55.31%	51.18%	61.57%	53.50%	51.56%	38.57%	59.65%	52.03%
GCLR [31]	70.98%	73.12%	55.78%	60.89%	68.56%	70.45%	60.87%	64.57%	64.05%	67.26%
AGNN [32]	72.34%	74.56%	55.89%	62.31%	76.45%	75.68%	70.45%	72.96%	68.78%	71.38%
MAML [11]	68.27%	72.49%	53.09%	59.30%	76.25%	82.43%	65.86%	70.18%	65.87%	71.10%
LEO [12]	70.47%	74.61%	56.79%	62.11%	65.76%	69.24%	59.16%	61.67%	63.04%	66.91%
**FRSM (Ours)**	67.62%	72.57%	55.03%	63.82%	**82.55%**	**88.93%**	**74.75%**	**79.85%**	**69.99%**	**76.29%**
Baseline [3]	47.96%	55.22%	42.67%	50.11%	42.00%	49.05%	24.73%	26.60%	39.34%	45.25%
Baseline++ [3]	43.72%	49.63%	36.92%	42.11%	41.11%	45.24%	23.44%	24.84%	36.30%	40.45%
SKD [23]	49.59%	57.05%	41.78%	49.23%	41.11%	47.10%	23.22%	25.56%	38.92%	44.74%
ProtoNet [10]	49.82%	56.08%	41.07%	47.13%	41.96%	48.64%	25.03%	26.52%	39.47%	44.59%
RelationNet [24]	42.26%	39.47%	36.11%	33.18%	32.63%	28.25%	23.23%	22.82%	33.56%	30.93%
GCLR [31]	49.23%	54.67%	39.87%	44.09%	35.68%	39.01%	23.45%	25.78%	37.06%	40.89%
AGNN [32]	50.45%	55.78%	40.78%	48.60%	23.45%	46.90%	24.10%	27.01%	34.70%	44.57%
MAML [11]	43.85%	48.82%	38.54%	43.09%	42.38%	46.18%	23.34%	23.78%	37.03%	40.47%
LEO [12]	48.32%	53.45%	38.86%	43.32%	36.22%	38.47%	22.94%	23.59%	36.59%	39.71%
**FRSM (Ours)**	**51.25%**	**58.80%**	**43.66%**	**51.04%**	**43.45%**	**49.45%**	**25.12%**	**27.28%**	**40.87%**	**46.64%**

## Data Availability

Data available in a publicly accessible repository.

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
