# Peer review of "Revisiting the Transferability of Few-Shot Image Classification: A Frequency Spectrum Perspective"

_entropy, 2024, doi:10.3390/e26060473_

Round 1

Reviewer 1 Report

Comments and Suggestions for Authors

This study explores the issue of performance decline in few-shot image classification (FSIC) from a frequency spectrum perspective, introducing the Frequency Spectrum Mask (FRSM) method to enhance transferability by mitigating the impact of non-causal frequencies, demonstrating significant improvements across nine datasets.

Additional Comments:

1) Why were parts (a) and (b) of Figure 3 not combined? Combining these could clarify the related details and enhance the figure’s explanatory power.

2) The baseline comparisons used in the study appear outdated. Updating these with more recent methodologies could strengthen the validation of the FRSM method’s effectiveness, such as:

[a] Graph Complemented Latent Representation for Few-Shot Image Classification. TMM '23

[b] Graph Neural Networks With Triple Attention for Few-Shot Learning. TMM '23

3) There is an error in the citation of LEO in Table 1. Please correct the reference to ensure accuracy and reliability of the citations.

Overall, the vision of the work is impressive, but the actual scope of work appears somewhat limited.

Reviewer 2 Report

Comments and Suggestions for Authors

The paper investigates the problem of few-shot classification and focuses on understanding the transferability of extracted features across datasets through a frequency-based analysis.

The idea is interesting and the results indeed validate the author's claims. Furthermore, the paper is well written and the use of the English language is overall correct.   

Some comments that could enhance the quality of the paper are the following

Reference to “LEO” in section 2.1 is missing

It is not clear how causality is introduced. How are causal and non-causal components identified, i.e., how do different components get assigned to either the “causal” or “non-causal” components?

What is the impact of the support sample set size?

In Section 3.1, the problem of few shot classification is described as a two-step process, i.e. (i) pre-training and (ii) test-tuning phase. One could argue that pre-training is an intelligent initialization of the model so all the effort is focused on the fine-tuning step and the subsequent evaluation. In that sense, the discussion “The pre-trained model undergoes re-learning based on the few labeled images S at each gradient step and subsequently undergoes testing on the unlabeled images Q.” lacks sufficient detail.

The following text appears to be missing something “Because zAi represents the magnitude of the frequency.”

The discussion on how Fig 1 was created should be expanded by including information such the type of layer that was considered as input to the t-SNE, the feature dimensionality, number of training examples, and stability of performance as some indicative examples.

In Section 1, it is stated that “Figure 2 showcases the average 48

amplitudes of eigenfrequencies across four testing datasets using a pre-trained model”

Please provide more details regarding the characteristics of the actual signal where the eigendecomposition takes place, i.e., is the eigenanalysis applied

Provide results on the following assertion “For example, consider a scenario where the dog 55

images in the training data feature grass backgrounds, this scenario poses a challenge 56

during the testing phase when encountering the dog image with a water background, as 57

the inconsistent background information hampers recognition.”

Comments on the Quality of English Language

use of the English language is overall correct
